# Unravelling the Impact of Cyclic Mechanical Stretch in Keratoconus—A Transcriptomic Profiling Study

**DOI:** 10.3390/ijms24087437

**Published:** 2023-04-18

**Authors:** Theresa Akoto, Jingwen Cai, Sarah Nicholas, Hayden McCord, Amy J. Estes, Hongyan Xu, Dimitrios Karamichos, Yutao Liu

**Affiliations:** 1Department of Cellular Biology & Anatomy, Medical College of Georgia, Augusta University, Augusta, GA 30912, USA; 2North Texas Eye Research Institute, University of North Texas Health Science Center, Fort Worth, TX 76107, USA; 3Department of Pharmaceutical Sciences, University of North Texas Health Science Center, Fort Worth, TX 76107, USA; 4Department of Ophthalmology, Augusta University, Augusta, GA 30912, USA; 5James & Jean Culver Vision Discovery Institute, Medical College of Georgia, Augusta University, Augusta, GA 30912, USA; 6Department of Population Health Sciences, Medical College of Georgia, Augusta University, Augusta, GA 30912, USA; 7Department of Pharmacology and Neuroscience, University of North Texas Health Science Center, Fort Worth, TX 76107, USA; 8Center for Biotechnology and Genomic Medicine, Medical College of Georgia, Augusta University, Augusta, GA 30912, USA

**Keywords:** cornea, keratoconus (KC), cyclic mechanical stretch (CMS), TGFβ1 treatment, RNA-Sequencing

## Abstract

Biomechanical and molecular stresses may contribute to the pathogenesis of keratoconus (KC). We aimed to profile the transcriptomic changes in healthy primary human corneal (HCF) and KC-derived cells (HKC) combined with TGFβ1 treatment and cyclic mechanical stretch (CMS), mimicking the pathophysiological condition in KC. HCFs (n = 4) and HKCs (n = 4) were cultured in flexible-bottom collagen-coated 6-well plates treated with 0, 5, and 10 ng/mL of TGFβ1 with or without 15% CMS (1 cycle/s, 24 h) using a computer-controlled Flexcell FX-6000T Tension system. We used stranded total RNA-Seq to profile expression changes in 48 HCF/HKC samples (100 bp PE, 70–90 million reads per sample), followed by bioinformatics analysis using an established pipeline with Partek Flow software. A multi-factor ANOVA model, including KC, TGFβ1 treatment, and CMS, was used to identify differentially expressed genes (DEGs, |fold change| ≥ 1.5, FDR ≤ 0.1, CPM ≥ 10 in ≥1 sample) in HKCs (n = 24) vs. HCFs (n = 24) and those responsive to TGFβ1 and/or CMS. PANTHER classification system and the DAVID bioinformatics resources were used to identify significantly enriched pathways (FDR ≤ 0.05). Using multi-factorial ANOVA analyses, 479 DEGs were identified in HKCs vs. HCFs including TGFβ1 treatment and CMS as cofactors. Among these DEGs, 199 KC-altered genes were responsive to TGFβ1, thirteen were responsive to CMS, and six were responsive to TGFβ1 and CMS. Pathway analyses using PANTHER and DAVID indicated the enrichment of genes involved in numerous KC-relevant functions, including but not limited to degradation of extracellular matrix, inflammatory response, apoptotic processes, WNT signaling, collagen fibril organization, and cytoskeletal structure organization. TGFβ1-responsive KC DEGs were also enriched in these. CMS-responsive KC-altered genes such as *OBSCN*, *CLU*, *HDAC5*, *AK4*, *ITGA10*, and *F2RL1* were identified. Some KC-altered genes, such as *CLU* and *F2RL1*, were identified to be responsive to both TGFβ1 and CMS. For the first time, our multi-factorial RNA-Seq study has identified many KC-relevant genes and pathways in HKCs with TGFβ1 treatment under CMS, suggesting a potential role of TGFβ1 and biomechanical stretch in KC development.

## 1. Introduction

Keratoconus (KC) is one of the most common corneal ectasia and a leading indicator of corneal transplantation worldwide [1,2,3,4]. It’s defined by the gradual thinning and steepening of the typical dome-shaped cornea into a cone-shaped form, resulting in intensified irregular astigmatism, myopia, increased corneal sensitivity, and a decrease in visual acuity [5,6,7]. It generally begins in adolescence, lasts 10–20 years, and then stabilizes in the third or fourth decade of a person’s life [8,9]. It affects both males and females and all ethnicities with varied incidence rates among various ethnic groups and geographical locations—4.79% out of 522 patients in Saudi Arabia [10], 2.3% out of 4711 subjects in Central India [11], 2.34% out of 981 volunteers in Jerusalem [12], 3.3% out of 92 students in Lebanon [13,14], and 2.5% out of 1027 students in Iran [15]. The reported variation in KC observed amongst the various ethnic groups may be due to a number of factors including genetics, environmental variables, study design, age of participants, clinical exams, and diagnostic criteria [14,16,17]. Although poorly understood, previous studies have proposed the association of KC with several environmental factors such as ultraviolet radiation exposure [18,19], the frequent wearing of contact lenses [20,21], abnormal eye rubbing [22,23,24], genetic factors [25,26], and hormonal imbalance [27,28].

Numerous studies have reported the TGFβ signaling pathway as a regulator of ECM gene expression [29,30,31]. It has been shown to play roles in several physiological and pathological conditions such as cell proliferation and differentiation, carcinogenesis, autoimmunity, angiogenesis, apoptosis, and wound healing [32,33,34,35]. TGFβ is a multifunctional cytokine with three isoforms (TGFβ1, TGFβ2 (both pro-fibrotic), and TGFβ3 (anti-fibrotic)), that bind to the TGFβ receptors with all three isoforms expressed in the human cornea [29,36,37]. Upon corneal injury, TGFβ1 binds to TGFβ receptor II (TGFBR2) and increases the production of connective tissue growth factor (CTGF), which in turn activates keratocytes to produce collagen [38]. In KC, studies have shown TGFβ1 as a key modulator of ECM assembly [39]. Primary human corneal KC stromal fibroblast cells exhibit a myofibroblastic phenotype which contributes to fibrosis and altered ECM assembly compared to non-KC human corneal stromal fibroblast cells [39,40]. Further, treatment of HKCs with TGFβ1 induces the expression of matrix metallopeptidase-1 and -3 (MMP-1 and MMP-3) compared to HCFs [41]. Clearly, altered TGFβ1 signaling may play a key role in KC pathogenesis. Thus, determining whole transcriptomic changes in response to TGFβ1 could be beneficial in understanding the different players that could be associated with the altered signaling cascade in KC.

Besides TGFβ as a molecular factor, biomechanical factors have also been shown to contribute to KC [42,43]. Due to the exposure of the healthy cornea to the external environment, it can be subjected to a variety of external influences (including changes in air pressure, eyelid motions, eye rubbing, dehydration, and so on) as well as internal factors such as daily intraocular pressure (IOP) swings [44]. The cornea is described as viscoelastic and hence prone to structural changes when exposed to these external forces and internal stresses [45]. One of the significant predictors of KC, according to a multivariate analysis of risk factors that may contribute to KC, is chronic extensive eye rubbing [46]. This observation has been shown in separate studies at different frequencies; 65.6% of 244 KC patients in a survey conducted had a history of eye rubbing [47]. Two independent observational studies reported approximately 50% of KC patients with frequent and vigorous eye rubbing [48,49]. Overall, eye rubbing is hard to quantify without encountering subjective bias.

It is believed that vigorous knuckle-grinding and repetitive moderate rubbing seen in KC patients may cause corneal deformations and significant pathological and biochemical alterations [46,50]. In addition, changes secondary to eye rubbing may include increased concentrations of inflammatory factors, MMP/tissue inhibitors of metalloproteinases (TIMPs) imbalance, keratocytes cell death, and scar formation, leading to ECM degradation and thinning of the corneal stroma [51]. Furthermore, investigations have shown that forceful eye rubbing, eye blinking, and ocular pulse in the cornea cause substantial variations in IOP [52,53,54,55]. These primary and secondary changes due to eye rubbing could make the cornea more susceptible to the effects of IOP, leading to more corneal thinning and stretching outward, forming the “cone” in KC. Therefore, we hypothesize that due to the dynamic forces applied to the cornea within the intraocular environment, KC may continuously progress. A defect in the structural integrity of the cornea owing to the biochemical changes renders it vulnerable to ECM remodeling and gene expression changes [56]. 

Several cytomechanical studies have applied mechanical stresses such as periodic stretching, pressure, shearing, or an electric field to corneal fibroblast cells, aiming to determine how the strength, duration, and frequency of these mechanical forces affect cellular responses [45,57,58]. However, most of these studies were more targeted and focused on looking at the expression of specific gene changes and pathways caused by one factor of KC. To the best of our knowledge, no report exists on the transcriptome changes due to molecular and biomechanical contributions in KC.

Gene expression profiling is a valuable tool to look at genes that are expressed in a cell at a given time to enhance the prediction of relevant pathways, which can provide insights into the understanding of diseases [59]. RNA-Seq is one of the powerful gene expression profiling tools which can be used to sequence transcriptome without prior knowledge [60,61].

Our study aimed to determine, for the first time, the transcriptomic changes in HCFs and HKCs under TGFβ1 treatment with cyclic mechanical stretch (CMS) to model the pathophysiological condition in KC using total RNA-Seq. We hypothesize that differentially expressed genes and pathways in HKCs with TGFβ1 treatment under CMS can contribute to our understanding of the molecular mechanisms in KC. Since KC is multifactorial and causes progressive vision loss, determining the combined effects of multiple factors that govern its development and progression might lead to new therapeutic options.

## 2. Results

### 2.1. Identification of Differentially Expressed Genes

Our study included four healthy HCF and four KC-derived cells treated with three different concentrations of TGFβ1 (0, 5, 10 ng/mL) with or without CMS, leading to a total of 48 samples. On average, an RNA integrity number (RIN) ≥ 8 was obtained for all 48 samples. RNA-Seq generated about 70–90 million 100 bp paired-end reads with an average read quality score > 35. Using a cutoff of CPM ≥ 10 in at least one sample, 11,155 genes were identified in all 48 samples (Appendix A). 

With an absolute fold change cutoff of ≥ 1.5 and FDR ≤ 0.1, our DE multi-factorial ANOVA with KC status, TGFβ1 treatment, and CMS status identified 479 DE genes (DEG) in HKCs vs. HCFs with TGFβ1 treatment and CMS status as cofactors (HKCs (TGFβ1 + CMS) vs. HCFs (TGFβ1 + CMS)) (Appendix A). Of these genes, 95 were upregulated, and 384 were downregulated (Appendix A). All these DE genes with their FDR values are shown in the volcano plot in Figure 1. Interestingly, some of these genes, such as *ADAMTS15*, *HDAC5*, *COL16A1*, *COL7A1*, *COL4A2*, *WNT2*, *LAMA3*, *ITGA10* and *AQP1* have been identified as KC-related genes or expressed in the cornea [38,62,63,64,65,66,67,68,69]. Listed in Table 1 are the top 10 upregulated and top 10 downregulated genes in HKCs vs. HCFs including TGFβ1 treatment and CMS as cofactors.

To examine the impact of the combined effect of TGFβ1 treatment and CMS on normal stromal fibroblasts, we compared the expression data between the HCFs with 10 ng/mL TGFβ1 treatment and CMS and HCFs without TGFβ1 treatment or CMS and identified a total of 748 DE genes with FDR < 0.1 and |FC| ≥ 1.5 in treated HCFs (Appendix A). Interestingly, 116 of these DE genes in treated HCFs were also differentially expressed in HKCs (Appendix A) identified through the multi-factorial ANOVA analyses (Figure 2). Listed in Table 2 are the six up- and 10 downregulated overlapping DE genes. The partial overlapping of DE genes in HCFs under combined treatment with HKCs suggests the potential contribution of these two KC-relevant factors (molecular and biomechanical factors) in KC pathobiology.

To identify the functional categories the DEGs from HKCs (TGFβ1 + CMS) vs. HCFs (TGFβ1 + CMS) were enriched in, the 479 DEGs were uploaded to the PANTHER classification system and the DAVID bioinformatics. Gene ontology analysis revealed that these genes were involved in biological processes such as the Interleukin-27-mediated signaling pathway, actomyosin structure organization, regulation of ERK1 and ERK2 cascade, regulation of actin filament depolymerization, eye morphogenesis, positive regulation of non-canonical WNT signaling, regulation of apoptosis process, collagen fibril organization, integrin-mediated signaling pathway and fibroblast migration (Figure 3a, Appendix A). In addition, these genes were enriched in molecular functions such as extracellular matrix structural constituent, extracellular matrix binding, calcium ion binding, glycosaminoglycan binding, actin binding, integrin binding, and Transporter associated with antigen processing (TAP) binding (Figure 3b, Appendix A). Furthermore, these genes were found to be localized in the interstitial matrix, basement membrane, endoplasmic reticulum lumen, microfibril, actin cytoskeleton and extracellular exosome (Figure 3c, Appendix A). PANTHER pathway analysis indicated the enrichment of genes involved in the WNT signaling pathway, integrin signaling pathway, and cadherin signaling pathway (Figure 4a, Appendix A). In addition, Reactome analysis suggested the enrichment of genes in the degradation of extracellular matrix, laminin interactions, defective human beta-1,3-glucosyltransferase like protein (B3GALTL) in Peters plus Syndrome (Pps), integrin cell surface interactions, interferon gamma signaling, interferon alpha/beta signaling, maturation of nucleoprotein, O-glycosylation of thrombospondin type 1 repeat (TSR) domain-containing proteins, and collagen biosynthesis and modifying enzymes (Figure 4b, Appendix A).

Among the total DEGs identified, 199 KC-altered genes were responsive to TGFβ1 treatment significantly (FDR < 0.1) (Appendix A). There was a dose-dependent increase (positive correlation) in the expression of 60 of these genes (Appendix A), whereas 139 genes were negatively correlated with TGFβ1 treatment (Appendix A). Listed in Table 3 are the top 10 DE genes with positive correlations and the top 10 DE genes with negative correlations with KC.

Gene ontology analysis with the 199 KC-altered genes responsive to TGFβ1 treatment revealed biological processes such as positive regulation of WNT signaling pathway, planar cell polarity pathway, protein mono-ADP-ribosylation, fibroblast migration, extracellular matrix organization, positive regulation of pattern recognition receptor signaling pathway, and response to type 1 interferon (Figure 5a, Appendix A). In addition, these genes were involved in molecular functions such as ECM structural constituent and protein binding (Figure 5b, Appendix A). Furthermore, the genes were found to be localized in the basement membrane, anchored components of the membrane, and extracellular space (Figure 5c, Appendix A). Reactome analysis revealed genes enriched in the extracellular matrix organization, interferon alpha/beta signaling, interferon signaling, and cytokine signaling in the immune system (Figure 6, Appendix A).

Thirteen KC-altered genes were identified to be responsive to CMS (Table 4), and six were responsive to TGFβ1 treatment and CMS together (Table 5).

### 2.2. Droplet Digital PCR (ddPCR) Validation of Differentially Expressed Genes

To validate the RNA-Seq data, we analyzed the differential expression of five selected genes (*COL7A1*, *SCARA3*, *SERPINF1, FBN2*, and *CLU*) in the same samples used for the RNA-Seq. These genes were selected based on their expression and contributions to the identified pathways. *COL7A1*, *SCARA3*, *SERPINF1*, and *FBN2* were validated from genes expressed in HKCs vs. HCFs with TGFβ1 treatment and CMS status as cofactors (Appendix A). *COL7A1* and *SERPINF1* were validated from KC-altered genes responsive to TGFβ1 treatment (Appendix A). *CLU* was validated from KC-altered genes responsive to TGFβ1 treatment and/or CMS only (Table 4 and Table 5). We compared our ddPCR findings with the RNA-Seq data for each group. For disease status (HKCs vs. HCFs with TGFβ1 treatment and CMS status as cofactors), *SERPINF1* and *FBN2* were downregulated, consistent with the RNA-Seq data (Table 6). However, the expression of *FBN2* failed to reach statistical significance (Table 6). The expression of *COL7A1* and *SCARA3* were upregulated with ddPCR, whereas they were downregulated in RNA-Seq (Table 6) which could be due to the subtle differences in the multi-factorial statistical models and data normalization between the RNA-Seq and ddPCR data. For KC-altered genes responsive to TGFβ1 treatment, we found a negative correlation of *SERPINF1* expression with TGFβ1 treatment, consistent with the RNA-Seq findings (Table 7). *COL7A1* showed a positive correlation with TGFβ1 treatment, similar to our RNA-Seq data (Table 7). However, its expression in HKCs showed an opposite trend to the RNA-Seq data (Table 7). Lastly, for KC-altered genes responsive to TGFβ1 treatment and/or CMS only, we observed downregulation of *CLU* expression similar to the RNA-Seq results. However, it did not reach statistical significance for its response to either CMS or TGFβ1 treatment (Table 8 and Table 9).

## 3. Discussion

### 3.1. Overview

For the first time, we determined the transcriptomic changes in HCFs and HKCs by combining more than one risk factor of KC, a biomechanical factor (CMS) and a molecular factor (TGFβ1 treatment) to model KC as a multifactorial disease. Using DE multifactorial ANOVA with KC status, TGFβ1 treatment, and CMS status, we identified 479 DE genes in HKCs (TGFβ1 + CMS) vs. HCFs (TGFβ1 + CMS), 199 KC-altered genes responsive to TGFβ1 treatment, 13 KC-altered genes responsive to CMS and 6 KC-altered genes responsive to TGFβ1 treatment and CMS together. We also compared the expression data between the HCFs with 10ng/mL TGFβ1 treatment and CMS and HCFs without TGFβ1 treatment or CMS to determine the impact of the combined effect of TGFβ1 treatment and CMS on stromal fibroblasts and identified a total of 748 DE genes in treated HCFs; 116 of these DE genes in treated HCFs were also differentially expressed in HKCs (TGFβ1 + CMS) vs. HCFs (TGFβ1 + CMS). Differentially expressed genes identified in HKCs (TGFβ1 + CMS) vs. HCFs (TGFβ1 + CMS) were enriched in several pathways such as WNT signaling, regulation of apoptosis process, collagen fibril organization, and degradation of extracellular matrix, all of which have been implicated in KC pathogenesis [70,71,72,73,74]. TGFβ1-responsive KC DEGs were also found to be enriched in these pathways. We identified KC-altered genes that were responsive to CMS and/or TGFβ1 treatment. Altogether, our study has highlighted some potential pathways and biomarkers and the need to understand how multiple factors could drive KC pathogenesis.

### 3.2. Differentially Expressed Genes in HKCs vs. HCFs and KC-Altered Genes Responsive to TGFβ1 Treatment

Some genes we identified, such as *SERPINF1* and *FBN2*, may play functional roles in the cornea or be involved with ECM [62,75]. *SERPINF1* encodes PEDF (pigment epithelium-derived growth factor). PEDF is a collagen-binding protein with high expression in many tissues such as bone, skeletal muscle, spleen, kidney, brain, and, interestingly, in the cornea [75,76]. It plays a role in many biological processes, such as the inhibition of angiogenesis, bone formation, and interaction with the components of ECM, such as glycosaminoglycans and collagens [77,78,79,80]. Among all the collagen family members, PEDF has been reported to have the highest affinity for collagen I [80]. Interestingly, Type I collagen comprises most of the corneal stroma’s collagen, with lower quantities of the other types [81]. Even though it has not been demonstrated that PEDF is expressed in the corneal stroma or in KC, it was markedly downregulated in HKCs. This finding suggests a possible alteration of collagen molecular assembly since its interaction with collagen may affect the corneal stroma of KC patients.

The TGFβ pathway is essential in controlling extracellular matrix gene expression [30]. There is an opposing relationship between TGFβ1 and PEDF in a study on type VI osteogenesis imperfecta pathogenesis using *Serpinf1* (−/−) mouse osteoblasts [82]. TGFβ1 treatment increased the expression of pro-angiogenic factor genes in *Serpinf1* (−/−) mouse osteoblasts [82]. In our study, a dose-dependent increase in TGFβ1 treatment reduced the expression of *SERPINF1*, indicating that TGFβ1 treatment may augment the alterations of collagen molecular assembly as aforementioned by reducing SERPINF1 expression.

Studies have also shown the interplay between TGFβ and some environmental factors of KC such as UV exposure and molecular factors such as the expression of MMPs [41,83]. For example, Quan et al., have shown that UV exposure affects the TGFβ/Smad pathway by altering the expression of the three TGFβ isoforms, their receptors, and downstream signaling genes such as *SMAD2*, *SMAD3*, *SMAD4*, and *SMAD7* in human skin in vivo [83]. In addition, UV exposure has been reported to decrease the synthesis of type I and type III procollagen in human skin in vivo [84]. In the human skin, TGFβ/Smad pathway acts as a key regulator of skin fibroblasts in the production of type I and type III collagen [83]. Given the important role of the TGFβ/Smad pathway, alteration of the pathway by UV exposure could result in a decrease in the synthesis of type I and type III procollagen. Interestingly, a decrease in collagen I, III, and V is evident in HKCs following corneal collagen crosslinking (CXL) using a 3-D in vitro CXL model [85]. CXL requires the use of UVA and riboflavin to enhance the biomechanical stability of the cornea by halting KC progression [39,86]. Sharif et al. observed a decrease in SMAD7 in HKCs after CXL [85]. SMAD7 negatively regulates TGFβ signaling by acting as an antagonist of the TGFβ family of proteins [87]. Thus, the downregulation of SMAD7 in HKCs after CXL confirms the aberrant TGFβ signaling pathway reported in KC [29,31]. UV exposure has also been shown to upregulate the expression of pro-inflammatory cytokines such as IL6, IL8, and tumor necrosis factor alpha (TNFα) in limbal fibroblasts [88], play a role in Pterygium by increasing the expression of cysteine rich transmembrane BMP regulator 1 (CRIM1) [89] and induce expressions of MMPs (MMP-14, MMP-2, MMP-1, MMP-9, MMP-2, MMP-7, MMP-8) in corneal epithelial cells [90].

Lyon et al. have also shown a connection between TGFβ treatment and MMP expression in HKCs and HCFs [41]. The authors observed a significant increase in MMP-1 and MMP-3 expression in HKCs compared to HCFs after treatment with 0.1 ng/mL of TGFβ1 [41]. With this, it can be speculated that a dose dependent increase in TGFβ1 from our study could enhance the expression of these MMPs in KC corneas. Altogether, the interrelationship between the different risk factors of KC reinforces the need to combine more than one risk factor of KC to understand the disease pathogenesis.

### 3.3. KC-Altered Genes Responsive to CMS

Mechanotransduction has been studied in various cell types and diseases, including vascular endothelial cells, trabecular meshwork cells, mesenchymal stem cells, scleral fibroblasts, glaucoma, ventilator-induced lung injury, and lamina cribosa cells using the in vitro cell stretch systems to determine the effects of mechanical forces on these cells [82,91,92,93,94]. In our transcriptomic study, we identified KC-altered genes responsive to CMS. Several genes—*CLU*, *AK4*, and *F2RL1* have been highlighted in cellular and biological processes, which may be directly or indirectly related to KC or cornea function. Adenylate kinase 4 (AK4) is a member of the nucleotide monophosphate kinase family, which plays a role in energy metabolism [95]. As a stress-response protein, AK4 is essential for cell survival by protecting cells in response to oxidative stress [96]. Increased levels of oxidative stress markers have been identified in KC corneas implying that oxidative stress could contribute to the pathogenesis of KC [97,98,99]. F2R Like Trypsin Receptor 1 (F2RL1), also known as Protease-activated receptor 2 (PAR2), has been shown to play a role in the inflammatory response [100]. In addition, studies have shown that activation of PAR2 induces the secretion of MMP-9 in human airways [100,101]. Interestingly, increased levels of MMP-9 and inflammatory cytokine expression have been observed in the tears of KC patients [102]. Although not reported in the human cornea, expression of F2RL1 in response to CMS could be a potential inducer of MMP-9 in KC corneas. Further studies may be needed to warrant this claim.

### 3.4. Limitations of Our Study

Our study has a few limitations. First, we used primary corneal stromal and KC fibroblast cells cultured in 10% fetal bovine serum (FBS) instead of primary keratocytes. This could affect their gene expression patterns. In the corneal stroma, keratocytes are quiescent, dendritic cells that are activated (corneal fibroblasts) upon injury [103]. In numerous cell culture studies, primary keratocytes isolated from corneal stromal tissue are maintained with 10% FBS to stimulate proliferation [104,105,106]. However, this can transform them into fibroblasts or myofibroblasts phenotypes. Reducing the concentration of FBS to about 1–2% has been shown to decrease the phenotype transition while maintaining a keratocyte-like phenotype [107,108]. Second, to achieve the CMS, our cells (both experimental and control) were cultured on flexible-bottom collagen-coated plates, which provided a softer substrate for cell growth, unlike the regular plastic culture dishes. Changes in substrate stiffness could affect cell behavior and modulate gene expression [109,110]. Third, the CMS was conducted at a single time point of 24 h with a 15% stretching regimen. Since vigorous eye rubbing is frequently seen in KC patients, performing time-series experiment with increasing stretching regimens could explain how the frequency and magnitude of the mechanical force exerted on the cornea affect molecular responses. Additionally, it should be noted that TGFβ1 inside the cornea is not released in the active form while our treatment used the active form of TGFβ1, which might drive gene expression into a pro- or autoinflammatory condition in a dose-dependent manner. Lastly, our small sample size could affect the incomplete validation of the RNA-Seq expression using ddPCR. The differences in age, sex, and ethnicity may also contribute to incomplete validation. The subtle differences in the multi-factorial statistical models in the RNA-Seq data and ddPCR could contribute to the different expression changes.

### 3.5. Conclusions

Since KC pathogenesis is multifactorial, it is necessary to integrate multiple contributing factors into KC-related cellular and molecular studies. For the first time, we have successfully profiled the transcriptomic changes in HCF and HKC cells by combining more than one risk factor of KC, a biomechanical factor (CMS) and a molecular factor (TGFβ1 treatment) to model the disease using RNA-Seq. Our data have pointed out several genes and pathways that are altered in response to these risk factors suggesting a potential role of TGFβ1 and biomechanical stretch in KC development. Further studies may be required to characterize the functions of some of these genes. This could provide more understanding and serve as potential biomarkers in treating KC.

## 4. Materials and Methods

### 4.1. Culture of Primary Human Corneal Fibroblasts (HCFs) and KC-Derived Cells (HKCs)

All studies were performed in accordance with Institutional Review Board (IRB) approval at the University of North Texas Health Science Center (protocol #2020-030) and at the University of Oklahoma Health Sciences center (protocols #3450, #10108). The study adhered to the tenets of the Declaration of Helsinki. Written consents were obtained from all individuals with KC before corneal transplantation. KC tissues were obtained from the Oklahoma Health Sciences Center at Dean McGee Eye Institute (Oklahoma City, OK, USA) and from Aarhus University Hospital (Aarhus, Denmark). To confirm KC status, all KC donors were examined comprehensively using Pentacam HR, refraction, and slit lamps. Healthy corneal tissues with no history of ocular or systemic diseases were obtained from the National Development and Research Institutes (NDRI). Primary human corneal stromal fibroblast cells were isolated from four healthy and four KC individuals, as previously described [108,109,111]. Briefly, after removing the epithelium and the endothelium layers, small tissue pieces were attached to sterile flasks with EMEM containing 10% FBS (Atlanta Biologicals, Flowery Branch, GA, USA) and antibiotic/antimycotics (Life Technologies, Grand Island, NY, USA) at 37 °C with 5% CO_2_ for 2–3 weeks until cells migrated onto the flask.

HCFs and HKCs (Table 10) were maintained in Eagle’s Minimum Essential Medium (EMEM) with 1.5 g/L sodium bicarbonate, non-essential amino acids, L-glutamine, and sodium pyruvate (Corning, AZ, USA) supplemented with 10% fetal bovine serum (Atlanta Biologicals, Flowery Branch, GA, USA), and 1% antibiotic-antimycotic (ThermoFisher Scientific, IL, USA) at 37 °C with 5% CO_2_ in a humidified incubator. Cells from the third to sixth passages were used in all experiments.

HCF (n = 4) and HKC (n = 4) cells were cultured in flexible-bottom collagen-coated 6-well plates (Flexcell International Corporation, Burlington, NC, USA) at an initial density of 1.4 × 105/well treated with 0, 5, and 10 ng/mL of TGFβ1 (ThermoFisher Scientific, IL, USA) with or without 15% CMS (1 cycle/s, half-Sine mode, 24 h) using a computer-controlled Flexcell FX-6000T Tension system (Flexcell International Corporation, Burlington, NC, USA). Cells plated on Bioflex^®^ plates under the same conditions but not subjected to stretch served as controls.

### 4.2. RNA Extraction, Quality Check, Sequencing and Analysis

We extracted total RNA from each well using the miRNeasy Mini Kit (Qiagen, CA, USA) following the recommended procedures from the manufacturer. We evaluated the RNA quality using a 2100 Bioanalyzer with RNA 6000 Pico Kit (Agilent, Santa Clara, CA, USA). We only used samples with an RNA Integrity Number (RIN) ≥ 8 for RNA sequencing (RNA-Seq). A total of 100 ng RNA per sample was used to generate the sequencing libraries using the Zymo-Seq Ribofree Total RNA-Seq Library kit (Zymo Research Corporation, Irvine, CA, USA). After evaluating the library quality/size with Agilent Bioanalyzer High Sensitivity DNA kit and quantification with Qubit DNA assays, sequencing libraries from the 48 samples were pooled and sequenced with paired-end 100 bp using an Illumina NovaSeq 6000 system with an S4 flow cell at the Sequencing and Genomic Technologies Shared Resource at Duke University Center for Genomic and Computation Biology (70–90 million reads per sample). We evaluated the sequencing data for quality controls using Partek Flow. We aligned all the reads to human genome build hg38 using STAR version 2.7.8a, quantified to hg38-GENCODE Genes—release version 38, and normalized using counts per million mapped reads (CPM) and addition of 1 × 10^−4^ for missing values. Differential expression (DE) analyses were conducted using a multi-factorial ANOVA model, including KC status, TGFβ1 treatment, and CMS status.

For DE analysis, we excluded genes if their expression (CPM) was < 10 in all 48 samples. With this, only genes with CPM ≥ 10 in at least one sample were included for further analysis. For differential expression analysis missing data (i.e., values < 0.001) for any sample were replaced with a value of 0.0001. There are multiple different groups in the studies, HKCs and HCFs with and without CMS under TGFβ1 treatment with varying concentrations in this study. Simple comparisons between any two groups could identify specific DE genes. However, these simple comparisons would not be able to identify the combined effect of CMS and TGFβ1 treatment in HKCs/HCFs. Therefore, we performed multi-factorial ANOVA analyses (CMS, TGFβ1 treatment, donor ID as cofactors) to identify DE genes in HKCs vs. HCFs (Appendix A). Among these DE genes, we also identified those significantly correlated with TGFβ1 treatment, those affected significantly by CMS, and those significantly correlated with TGFβ1 treatment and affected by CMS at the same time (|fold change| ≥ 1.5, false discovery rate (FDR) ≤ 0.1 and CPM ≥ 10 in ≥1 sample). In addition, we also compared the expression data between HCFs with 10 ng/mL TGFβ1 treatment and CMS and those HCFs without TGFβ1 treatment or CMS. To determine the biological processes, cellular components, molecular functions, and significantly enriched pathways, we uploaded the complete list of DEGs to the PANTHER (Protein ANalysis THrough Evolutionary Relationships) classification system [112,113] and The Database for Annotation, Visualization, and Integrated Discovery (DAVID) (FDR ≤ 0.05) [114,115].

### 4.3. Validation of Differentially Expressed Genes Using ddPCR

Approximately 400 ng total RNA was used for mRNA reverse transcription using the High-Capacity cDNA Reverse Transcription Kits (Applied Biosystems, Foster City, CA, USA). We used a QX200 droplet digital PCR (ddPCR) system (Bio-Rad, Hercules, CA, USA) with the predesigned EvaGreen-based ddPCR assays (Bio-Rad, Hercules, CA, USA) to validate five differentially expressed genes *COL7A1*, *SCARA3*, *SERPINF1*, *FBN2* and CLU (Appendix A). We used the expression of one reference gene *GAPDH* (Appendix A) to normalize the expression of these selected genes of interest in 48 samples. We performed regression analysis based on a linear mixed-effect model to account for the correlations from the same sample. The sample effect was considered random, and the effects of disease status, stretch condition, and TGFβ1 concentrations were considered fixed in the mixed-effect model. The R package lme4 was used for the analysis.

## Figures and Tables

**Figure 1 ijms-24-07437-f001:**
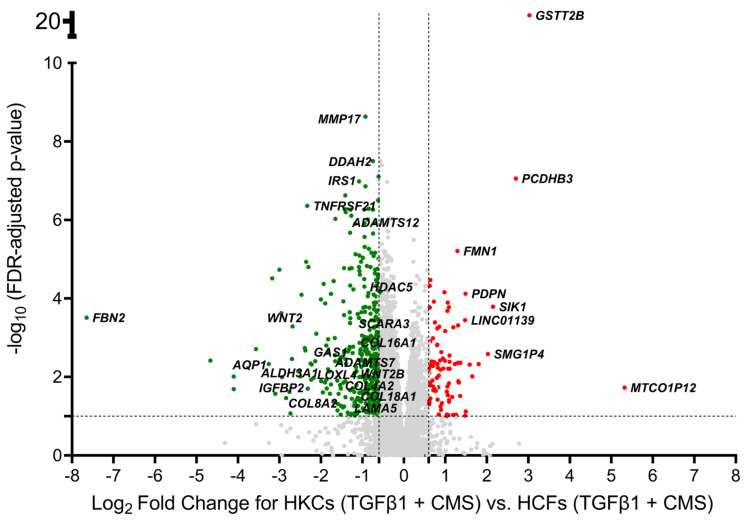
Genes that are differentially expressed in HKCs (TGFβ1 + CMS) vs. HCFs (TGFβ1 + CMS). Downregulated genes (FDR value ≤ 0.1, fold change ≤ −1.5, and CPM ≥ 10 in at least one sample) are labeled in GREEN. Upregulated genes (FDR value ≤ 0.1, fold change ≥ 1.5, and CPM ≥ 10 in at least one sample) are labeled in RED. Genes that are not significantly changed are labeled in GREY.

**Figure 2 ijms-24-07437-f002:**
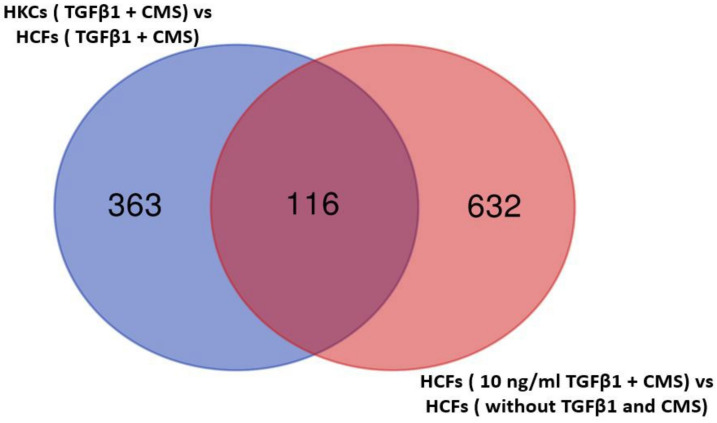
Four hundred and seventy-nine genes that are differentially expressed in HKCs (TGFβ1 + CMS) vs. HCFs (TGFβ1 + CMS) (FDR value ≤ 0.1, fold change ≥ 1.5, and CPM ≥ 10 in at least one sample) are labeled in BLUE. 748 DE genes in HCFs with and without 10 ng/mL TGFβ1 treatment and CMS (FDR value ≤ 0.1, fold change ≥ 1.5, and CPM ≥ 10 in at least one sample) are labeled in LIGHT RED; 116 represent overlapping DE genes in treated HCFs that were also differentially expressed in HKCs (TGFβ1 + CMS) vs. HCFs (TGFβ1 + CMS) (FDR value ≤ 0.1, fold change ≥ 1.5, and CPM ≥10 in at least one sample).

**Figure 3 ijms-24-07437-f003:**
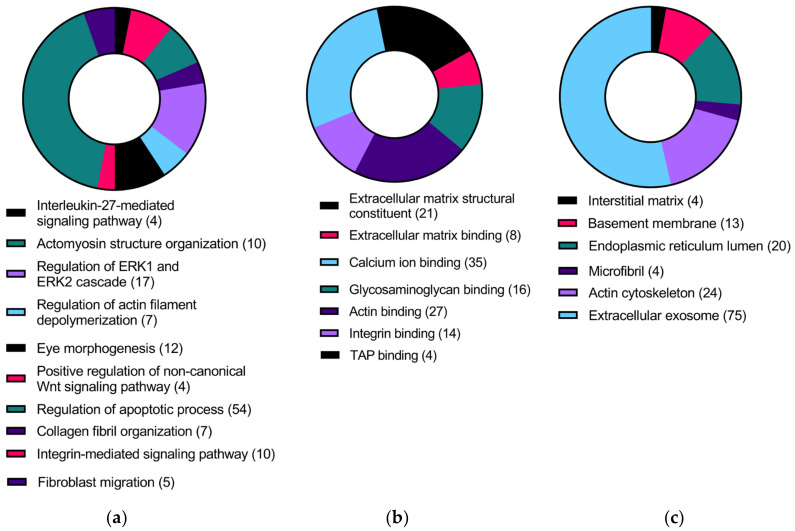
Gene ontology terms showing (**a**) biological processes, (**b**) molecular functions and (**c**) cellular components of significant differentially expressed genes in HKCs (TGFβ1 + CMS) vs. HCFs (TGFβ1 + CMS) (FDR ≤ 0.05). Numbers in brackets are the number of genes involved in each process.

**Figure 4 ijms-24-07437-f004:**
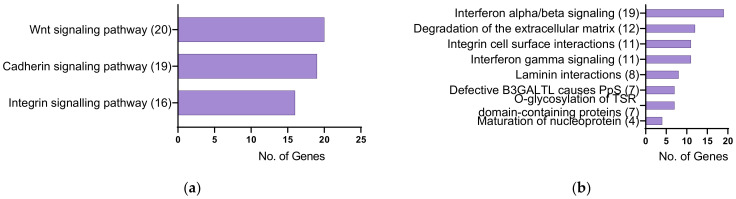
(**a**) PANTHER and (**b**) Reactome pathway analyses for significantly differentially expressed genes in HKCs (TGFβ1 + CMS) vs. HCFs (TGFβ1 + CMS) (FDR ≤ 0.05). Numbers in brackets are the number of genes involved in each pathway.

**Figure 5 ijms-24-07437-f005:**
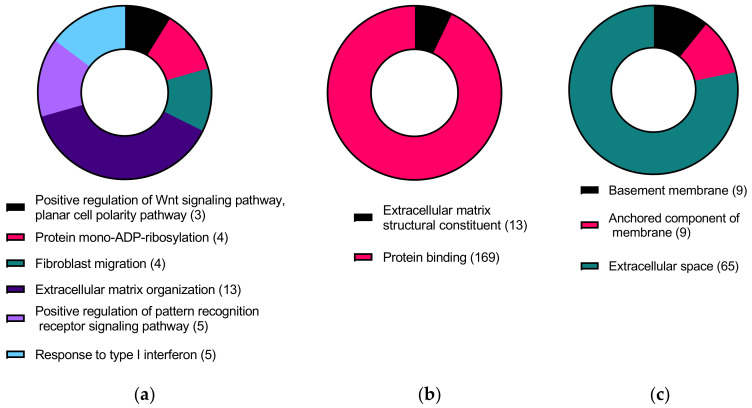
Gene ontology terms showing (**a**) biological processes, (**b**) molecular functions, and (**c**) cellular components of significant differentially expressed genes in KC-altered genes responsive to TGFβ1 treatment (FDR ≤ 0.05). Numbers in brackets are the number of genes involved in each process.

**Figure 6 ijms-24-07437-f006:**
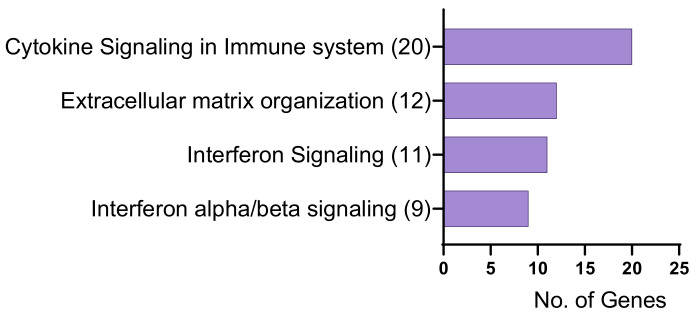
Reactome pathway analysis for significant differentially expressed genes in KC-altered genes responsive to TGFβ1 treatment (FDR ≤ 0.05) (FDR ≤ 0.05). Numbers in brackets are the number of genes involved in each pathway.

**Table 1 ijms-24-07437-t001:** Top 10 upregulated and 10 downregulated in HKCs (TGFβ1 + CMS) vs. HCFs (TGFβ1 + CMS).

Gene Symbol	Gene Description	Fold Change	FDR-Adjusted *p*-Value
Upregulated DEGs			
*MTCO1P12*	Mitochondrially Encoded Cytochrome C Oxidase I Pseudogene 12	39.9	1.87 × 10^−2^
*GSTT2B*	Glutathione S-Transferase Theta-2B	8.1	9.78 × 10^−24^
*PCDHB3*	Protocadherin Beta-3	6.5	8.83 × 10^−8^
*SIK1*	Salt Inducible Kinase 1	4.4	1.63 × 10^−4^
*SMG1P4*	SMG1 Pseudogene 4	4.1	2.62 × 10^−3^
*CDH10*	Cadherin 10	3.5	4.75 × 10^−3^
*RP11-649E7.5*	No description yet	3.1	9.74 × 10^−3^
*LAMC2*	Laminin Subunit Gamma 2	3.0	4.87 × 10^−3^
*NOTCH2NLB*	Notch 2 N-Terminal Like B	2.8	7.57 × 10^−2^
*PDPN*	Podoplanin	2.8	7.59 × 10^−5^
Downregulated DEGs			
*FBN2*	Fibrillin 2	−200.9	3.09 × 10^−4^
*SLC8A1*	Solute Carrier Family 8 Member A1	−25.4	3.83 × 10^−3^
*SORT1*	Sortilin 1	−17.2	9.82 × 10^−3^
*PKD1P2*	Polycystin 1, Transient Receptor Potential Channel Interacting Pseudogene 2	−17.2	2.03 × 10^−2^
*ITGA7*	Integrin Subunit Alpha 7	−11.8	1.96 × 10^−3^
*AQP1*	Aquaporin 1 (Colton blood group)	−9.6	4.69 × 10^−3^
*DUSP4*	Dual Specificity Phosphatase 1	−9.0	3.08 × 10^−5^
*ELN*	Elastin	−8.6	2.69 × 10^−2^
*OXTR*	Oxytocin Receptor	−8.0	1.85 × 10^−5^
*EPHB6*	EPH Receptor B6	−7.8	2.37 × 10^−4^

**Table 2 ijms-24-07437-t002:** Top 6 upregulated and 10 downregulated overlapping DE genes in HKCs (TGFβ1 + CMS) vs. HCFs (TGFβ1 + CMS) and HCFs (+/− 10 ng/mL TGFβ1 treatment and CMS).

Gene Symbol	Gene Description	HKCs (TGFβ1 + CMS) vs. HCFs (TGFβ1 + CMS)		HCF (+/− 10 ng/mL TGFβ1 and CMS)	
		Fold Change	FDR-Adjusted *p*-Value	Fold Change	FDR-Adjusted *p*-Value
Upregulated DEGs					
*LIPG*	Lipase G, Endothelial Type	2.1	1.46 × 10^−2^	9.4	4.27 × 10^−5^
*LAMC2*	Laminin Subunit Gamma 2	3.0	4.87 × 10^−3^	4.5	2.55 × 10^−6^
*MYO1D*	Myosin 1D	1.7	1.87 × 10^−2^	3.4	1.72 × 10^−2^
*BHLHE40*	Basic Helix-Loop-Helix Family Member E40	1.7	1.55 × 10^−3^	3.1	2.67 × 10^−9^
*SYT15*	Synaptotagmin 15	1.7	2.27 × 10^−2^	2.7	5.11 × 10^−2^
*CSMD2*	CUB And Sushi Multiple Domains 2	1.6	1.64 × 10^−2^	2.5	3.41 × 10^−2^
Downregulated DEGs					
*NPTX1*	Neuronal pentraxin 1	−2.0	1.01 × 10^−2^	−10.1	1.44 × 10^−14^
*CEMIP*	Cell migration inducing hyaluronidase 1	−2.3	4.52 × 10^−2^	−8.7	2.27 × 10^−26^
*IFI35*	Interferon induced protein 35	−1.7	3.01 × 10^−2^	−8.0	5.52 × 10^−5^
*SECTM1*	Secreted And Transmembrane 1	−3.7	1.59 × 10^−3^	−7.9	1.10 × 10^−4^
*CLDN11*	Claudin 11	−1.6	4.78 × 10^−3^	−5.3	8.50 × 10^−19^
*KCNJ2*	Potassium inwardly rectifying channel subfamily J member 2	−1.6	5.40 × 10^−2^	−4.8	1.65 × 10^−6^
*ALDH3A1*	Aldehyde Dehydrogenase 3 Family Member A1	−3.9	2.47 × 10^−2^	−4.7	5.49 × 10^−5^
*PLPP3*	Phospholipid Phosphatase 3	−1.6	4.01 × 10^−2^	−4.6	4.44 × 10^−17^
*GREM2*	Gremlin 2, DAN family BMP antagonist	−2.0	4.97 × 10^−3^	−4.5	2.42 × 10^−11^
*GJD3*	Gap Junction Protein Delta 3	−2.1	2.54 × 10^−2^	−4.3	5.06 × 10^−2^

**Table 3 ijms-24-07437-t003:** KC-altered genes with a positive correlation (top 10 genes) and a negative correlation (top 10 genes) with TGFβ1 treatment.

Gene Symbol	Gene Description/Homology	Fold Change	FDR-Adjusted *p*-Value (HKC vs. HCF)	Correlation (Conc (ng/mL))	FDR-Adjusted *p*-Value (Conc (ng/mL))
Positive correlation					
*SEMA7A*	Semaphorin 7A (John Milton Hagen Blood Group)	−1.6	3.74 × 10^−2^	0.7	1.19 × 10^−5^
*TENM4*	Teneurin Transmembrane Protein 4	−1.6	1.14 × 10^−2^	0.7	1.95 × 10^−5^
*PSD4*	Pleckstrin and Sec7 Domain Containing 4	−2.4	5.46 × 10^−7^	0.7	3.97 × 10^−5^
*POU2F2*	POU Class 2 Homeobox 2	−1.9	2.71 × 10^−6^	0.6	9.02 × 10^−5^
*COL7A1*	Collagen Type VII Alpha 1 Chain	−2.0	1.59 × 10^−3^	0.6	9.31 × 10^−5^
*ZNF365*	Zinc Finger Protein 365	−3.2	1.02 × 10^−3^	0.6	9.88 × 10^−5^
*CNN1*	Calponin 1	−2.5	1.93 × 10^−3^	0.6	9.88 × 10^−5^
*UCN2*	Urocortin 2	−2.0	3.58 × 10^−3^	0.6	1.11 × 10^−4^
*PFKFB4*	6-Phosphofructo-2-Kinase/Fructose-2,6-Biphosphatase 4	−1.9	1.91 × 10^−3^	0.6	1.59 × 10^−4^
*BTBD11*	Homo sapiens KBTBD11 antisense RNA 1 (KBTBD11-AS1)	−1.8	6.40 × 10^−4^	0.6	2.22 × 10^−4^
Negative correlation					
*SSH3*	Slingshot Protein Phosphatase 3	−1.5	2.92 × 10^−4^	−0.8	1.35 × 10^−6^
*IFI35*	Interferon Induced Protein 35	−1.7	3.01 × 10^−2^	−0.7	2.63 × 10^−6^
*CLDN11*	Claudin 11	−1.6	4.78 × 10^−3^	−0.7	2.83 × 10^−6^
*BCAM*	Basal Cell Adhesion Molecule (Lutheran Blood Group)	−2.1	8.10 × 10^−6^	−0.7	1.26 × 10^−5^
*CCN3*	Cellular Communication Network Factor 3	1.7	1.06 × 10^−2^	−0.7	1.64 × 10^−5^
*CD248*	CD248 Molecule	−1.5	1.47 × 10^−3^	−0.7	1.72 × 10^−5^
*PLPP3*	Phospholipid Phosphatase 3	−1.6	4.01 × 10^−2^	−0.7	2.25 × 10^−5^
*SECTM1*	Secreted and Transmembrane 1	−3.7	1.59 × 10^−3^	−0.7	2.76 × 10^−5^
*AHRR*	Aryl Hydrocarbon Receptor Repressor	−1.9	3.99 × 10^−4^	−0.7	5.36 × 10^−5^
*GREM2*	Gremlin 2, DAN Family BMP Antagonist	−2.0	4.97 × 10^−3^	−0.7	5.74 × 10^−5^

**Table 4 ijms-24-07437-t004:** KC-altered genes responsive to CMS.

Gene Symbol	Gene Description or Homology	Fold Change	FDR-Adjusted *p*-Value (HKC vs. HCF)	FDR-Adjusted *p*-Value (Stretch)
*DDAH2*	Dimethylarginine Dimethylaminohydrolase 2	−1.7	3.17 × 10^−8^	1.96 × 10^−2^
*OBSCN*	Obscurin, Cytoskeletal Calmodulin And Titin-Interacting RhoGEF	−2.7	2.37 × 10^−7^	7.36 × 10^−2^
*CLU*	Clusterin	−1.9	1.49 × 10^−5^	4.53 × 10^−2^
*LINC00565*	Homo sapiens chromosome 13 open reading frame 46 (C13orf46)	−1.6	1.86 × 10^−5^	2.30 × 10^−2^
*RP11-1334A24.5*	No description yet	−1.9	3.20 × 10^−5^	1.92 × 10^−2^
*HDAC5*	Histone Deacetylase 5	−1.5	6.45 × 10^−5^	9.94 × 10^−3^
*AK4*	Adenylate Kinase 4	1.7	5.78 × 10^−4^	1.12 × 10^−2^
*GSN-AS1*	GSN Antisense RNA 1	−1.7	9.95 × 10^−4^	9.65 × 10^−2^
*RPLP0P2*	Ribosomal Protein Lateral Stalk Subunit P0 Pseudogene 2	−2.1	1.36 × 10^−3^	7.98 × 10^−2^
*ITGA10*	Integrin Subunit Alpha 10	−2.1	2.12 × 10^−3^	1.84 × 10^−3^
*KCNS3*	Potassium Voltage-Gated Channel Modifier Subfamily S Member 3	−1.5	2.33 × 10^−3^	5.18 × 10^−2^
*CLDN11*	Claudin 11	−1.6	4.78 × 10^−3^	4.04 × 10^−2^
*F2RL1*	F2R Like Trypsin Receptor 1	−2.2	7.18 × 10^−3^	6.15 × 10^−2^

**Table 5 ijms-24-07437-t005:** KC-altered genes responsive to TGFβ1 treatment and CMS.

Gene Symbol	Gene Description/Homology	Fold Change	FDR-Adjusted *p*-Value (HKC vs. HCF)	Correlation (Conc (ng/mL))	FDR-Adjusted *p*-Value (Conc (ng/mL))	FDR-Adjusted *p*-Value (Stretch)
*DDAH2*	Dimethylarginine Dimethylaminohydrolase 2	−1.7	3.17 × 10^−8^	−0.4	3.49 × 10^−2^	1.96 × 10^−2^
*CLU*	Clusterin	−1.9	1.49 × 10^−5^	−0.2	3.75 × 10^−3^	4.53 × 10^−2^
*LINC00565*	Homo sapiens chromosome 13 open reading frame 46 (C13orf46)	−1.6	1.86 × 10^−5^	0.5	5.64 × 10^−4^	2.30 × 10^−2^
*RP11-1334A24.5*	No description yet	−1.9	3.20 × 10^−5^	0.6	1.02 × 10^−3^	1.92 × 10^−2^
*CLDN11*	Claudin 11	−1.6	4.78 × 10^−3^	0.6	2.83 × 10^−6^	4.04 × 10^−2^
*F2RL1*	F2R Like Trypsin Receptor 1	−2.2	7.18 × 10^−3^	−0.1	1.00 × 10^−2^	6.15 × 10^−2^

**Table 6 ijms-24-07437-t006:** Validated differentially expressed genes with ddPCR compared with RNA-Seq for Disease status—HKCs (TGFβ1 + CMS) vs. HCFs (TGFβ1 + CMS).

Gene Symbol	Gene Name	ddPCR		RNA-Seq	
		Effect Size (HKCs vs. HCFs)	*p*-Value (HKCs vs. HCFs)	Fold Change (HKCs vs. HCFs)	FDR-Adjusted *p*-Value (HKCs vs. HCFs)
*COL7A1*	Collagen type VII alpha 1 chain	11.7	6.39 × 10^−4^	−2.0	1.59 × 10^−3^
*SCARA3*	Scavenger receptor class A member 3	18.3	7.57 × 10^−6^	−2.0	2.62 × 10^−3^
*SERPINF1*	Serpin family F member 1	−23.4	4.90 × 10^−4^	−2.4	3.19 × 10^−4^
*FBN2*	Fibrillin 2	−4.2	1.28 × 10^−1^	−200.9	3.09 × 10^−4^

**Table 7 ijms-24-07437-t007:** Validated differentially expressed genes with ddPCR compared with RNA-Seq for KC-altered genes with TGFβ1 treatment.

Gene Symbol	Gene Name	ddPCR				RNA-Seq			
		Effect Size (HKCs vs. HCFs)	*p*-Value (HKCs vs. HCFs)	Effect Size (TGFβ1 Treatment)	*p*-Value (TGFβ1 Treatment)	Fold Change (HKCs vs. HCFs)	FDR- Adjusted *p*-Value (HKCs vs. HCFs)	Correlation (Conc (ng/mL))	FDR- Adjusted *p*-Value (TGFβ1 Treatment)
*COL7A1*	Collagen type VII alpha 1 chain	11.7	6.39 × 10^−4^	1.8	3.75 × 10^−5^	−2.0	1.59 × 10^−3^	0.6	9.31 × 10^−5^
*SERPINF1*	Serpin family F member 1	−23.4	4.90 × 10^−4^	−2.3	7.21 × 10^−3^	−2.0	3.19 × 10^−4^	−0.4	1.64 × 10^−2^

**Table 8 ijms-24-07437-t008:** Validated differentially expressed genes with ddPCR compared with RNA-Seq for KC-altered genes responsive to CMS.

Gene Symbol	Gene Name	ddPCR			RNA-Seq		
		Effect Size (HKCs vs. HCFs)	*p*-Value (HKCs vs. HCFs)	*p*-Value (Stretch)	Fold Change (HKCs vs. HCFs)	FDR-Adjusted *p*-Value (HKCs vs. HCFs)	FDR-Adjusted *p*-Value (Stretch)
*CLU*	Clusterin	−5.4	0.1695	0.0952	−1.9	1.49 × 10^−5^	4.53 × 10^−2^

**Table 9 ijms-24-07437-t009:** Validated differentially expressed genes with ddPCR compared with RNA-Seq for KC-altered genes responsive to TGFβ1 treatment and CMS.

Gene Symbol	Gene Name	ddPCR					RNA-Seq				
		Fold Change (HKCs vs. HCFs)	*p*-Value (HKCs vs. HCFs)	Effect Size (TGFβ1 Treatment)	*p*-Value (TGFβ1 Treatment)	*p*-Value (Stretch)	Fold Change (HKCs vs. HCFs)	FDR-Adjusted *p*-Value (HKCs vs. HCFs)	Correlation (Conc (ng/mL))	FDR- Adjusted *p*-Value (TGFβ1 Treatment)	FDR- Adjusted *p*-Value (Stretch)
*CLU*	Clusterin	−5.4	0.1695	0.8	0.0979	0.0952	−1.9	1.49 × 10^−5^	−0.2	3.75 × 10^−3^	4.53 × 10^−2^

**Table 10 ijms-24-07437-t010:** Clinical phenotypes of primary human corneal stromal fibroblast cells derived from unaffected and KC patients.

Sample ID	Age (y/o)	Gender	Ethnicity	Cause of Death
HCF-1	65	Male	Caucasian	Cardiopulmonary Arrest (CPA)
HCF-2	63	Male	Caucasian	Acute STEMI
HCF-3	72	Female	Caucasian	Motor vehicle accident (MVA)
HCF-4	71	Male	Caucasian	Cardiorespiratory Failure
HKC-1	43	Female	Unknown	N/A (transplant patient)
HKC-2	19	Male	Unknown	N/A (transplant patient)
HKC-3	69	Male	Unknown	N/A (transplant patient)
HKC-4	31	Male	African American	N/A (transplant patient)

## Data Availability

The data presented in this report are included in the published article. All the data can be shared upon reasonable request by email.

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
