# Peer review of "Unravelling the Impact of Cyclic Mechanical Stretch in Keratoconus—A Transcriptomic Profiling Study"

_ijms, 2023, doi:10.3390/ijms24087437_

Round 1

Reviewer 1 Report

In the current manuscript, the authors studied the impact of cyclic mechanical stretch in keratoconus by using a potent transcriptomic profiling study. 

The study design lacks rigor. Since the combination of both TGFβ1 treatment and CMS as cofactors are supposed causes that lead to the development of KC; one would expect the combination of both in HCF to induce alterations in specific genes that will resemble the pathological program showed by HKC. That means authors should compare HCF vs. HKC alone to identify the different profiles in corneas from keratoconus vs. healthy subjects. Later the addition of TGFβ1 treatment and CMS to HCF should induce the program observed in HKC, to demonstrate their hypothesis.

Additionally, in the cornea, TGFB1 is released as an inactive precursor bound to its latency-associated peptide (LAP). This latent complex and latent TGFβ binding proteins (LTBP) bind and form the large latent complex. LTBP and LAP need later to be enzymatically cleaved by thrombospondin, matrix metalloproteinases (MMP) 2 and 9, reactive oxygen species, and different members of the integrin family such as αvβ6 or αvβ8 to activate TGFβ. This mechanism is different ex vivo where TGBB1 is added as an active form and can drive gene expression into pro or autoinflammatory depending in a dose-dependent manner.

Authors must specify clearly in the methods section, which is not described at all, HKC vs. HCF alone, with and without CMS and/or TGFB1. Additionally, since in the abstract authors describe different concentrations of TGFB1, these must be clearly described. From all those multiple groups, the authors should justify why they are only presenting two of them.

Most of the information provided in the introduction section is irrelevant to the study design and it is well-known. Authors should focus on what is unknown and provide a succinct rationale.

Some minor comments, but many others not included in the current revision, indicate that the corresponding authors did not pay enough attention to the manuscript:

“Increased from 1:2000 in the 1980s to 1:350 in the 56 2000s perhaps due to improved technological tools”. The 2020s are already 23 years ago, and more updated bibliophagy is expected. What is the relation between KC's increased prevalence and improved technological tools? What does it mean improved technological tools?

“Myriads of molecular factors such as increased levels in 62 metal metalloproteinases (MMPs) - MMP-1,9, and 13 …” It is important to emphasize to call the attention of the reviewer, but when the authors state “myriads” but they focus only in one (TGFB1), the work loss enthusiasm.

“The involvement of the TGFβ pathway in the regulation and synthesis 71 of ECM implies a role in KC pathogenesis, either in a causative function or a subsequent 72 repair response causing structural alterations in KC [36–38]”. This sentence is a statement; therefore, why is needed the current study?

“One of the significant predictors of KC, according to a multivariate analysis 80 of risk factors that may contribute to KC, is chronic extensive eye rubbing”. That is correct but still unknown whether patients rub the eye because they have KC, or they develop KC because they rub the eye. Is the purpose of CMS to mimic this behavior? If so, to initiate or to keep it chronic?

Author Response

Dear reviewer,

Reviewer 2 Report

The work done in “Unravelling the impact of cyclic mechanical stretch in keratoconus – A transcriptomic profiling study” is interesting for this field and clear to read. However, I have some comments that needs to be addressed:

Please discuss more about the other important keratoconus causative factor, the ultraviolet (UV) radiation that has been cited in the introduction. It is known that UV alters TGFb response in skin1, please consider previously described cellular mechanisms induced by UV in cornea, by citing works on inflammatory cytokines2, genetic associated factors3 and matrix metalloproteinases4; and discuss if and how these could be related to the ones that you have found herein.

It is not clear from the main text if the ddPCR has been done on the same samples that had been used for the RNA seq analysis. Please specify it at line 210. How can you explain the following results “Expression of COL7A1 and SCARA3 were upregulated with ddPCR, whereas they were downregulated in RNA-Seq“?

Minor revisions:

Line 52-53: substitute the second “increased” with “intensified” to avoid a repetition

Line 250: avoid repeating responsive

Figure 3a: increase characters size in the y axis labelling to be more consistent with the others

1             Quan, T., He, T., Kang, S., Voorhees, J. J. & Fisher, G. J. Ultraviolet irradiation alters transforming growth factor beta/smad pathway in human skin in vivo. The Journal of investigative dermatology 119, 499-506, doi:10.1046/j.1523-1747.2002.01834.x (2002).

2             Notara, M. et al. Short-term uvb-irradiation leads to putative limbal stem cell damage and niche cell-mediated upregulation of macrophage recruiting cytokines. Stem cell research 15, 643-654, doi:10.1016/j.scr.2015.10.008 (2015).

3             Maurizi, E. et al. A novel role for CRIM1 in the corneal response to UV and pterygium development. Experimental eye research 179, 75-92 (2019).

4             Ardan, T. & Cejková, J. Immunohistochemical expression of matrix metalloproteinases in the rabbit corneal epithelium upon UVA and UVB irradiation. Acta histochemica 114, 540-546, doi:10.1016/j.acthis.2011.10.004 (2012).

Author Response

Dear reviewer,
